# Early Detection and Dynamic Changes of Circulating Tumor Cells in Transgenic NeuN Transgenic (NTTg) Mice with Spontaneous Breast Tumor Development

**DOI:** 10.3390/cancers13133294

**Published:** 2021-06-30

**Authors:** Wen-Sy Tsai, Tsung-Fu Hung, Jia-Yang Chen, Shu-Huan Huang, Ying-Chih Chang

**Affiliations:** 1Division of Colon and Rectal Surgery, Department of Surgery, Chang Gung Memorial Hospital, College of Medicine, Chang Gung University, Taoyuan 33305, Taiwan; wensyt@gmail.com (W.-S.T.); Shhuang55@gmail.com (S.-H.H.); 2Genomics Research Center, Academia Sinica, Nankang, Taipei 115, Taiwan; roachbug@gmail.com (T.-F.H.); daifans@gmail.com (J.-Y.C.); 3National Laboratory Animal Center, National Research Laboratories, Taipei 115, Taiwan; 4Department of Chemical Engineering, Stanford University, Stanford, CA 94305, USA; 5Biomedical Translational Research Center, Academia Sinica, Taipei 115, Taiwan

**Keywords:** circulating tumor cells, murine model, early cancer detection, HER2/neu breast cancer

## Abstract

**Simple Summary:**

This study aimed to prove the early presence of circulating tumor cells (CTCs) with viability and tumorigenesis in a murine model that spontaneously develops breast cancer. Serial CTC examinations were performed on NeuN transgenic mice, starting from the age of 8 weeks and continuing after palpable tumor formation. Prior to the detection of palpable tumors, the CTC counts rose over time from 1 ± 1.6 to 16 ± 9.5 per 75 μL; this number continued to grow with tumor development. The viability and tumorigenesis of the collected CTCs were confirmed by re-implanting the cells into a non-cancer-bearing mouse. Ultrasonography with Doppler showed a significant correlation between CTCs and tumor vascular density (*p*-value < 0.01), rather than tumor volume (*p*-value 0.076).

**Abstract:**

Background: This study used NeuN transgenic (NTTg) mice with spontaneous breast tumor development to evaluate the dynamic changes of circulating tumor cells (CTCs) prior to and during tumor development. Methods: In this longitudinal, clinically uninterrupted study, we collected 75 μL of peripheral blood at the age of 8, 12, 16, and 20 weeks in the first group of five mice, and at the age of 32 weeks, the time of tumor palpability, and one week after tumor palpability in the second group of four mice. Diluted blood samples were run through a modified mouse-CMx chip to isolate the CTCs. Results: The CTC counts of the first group of mice were low (1 ± 1.6) initially. The average CTC counts were 16 ± 9.5, 29.0 ± 18.2, and 70.0 ± 30.3 cells per 75 μL blood at the age of 32 weeks, the time of tumor palpability, and one week after tumor palpability, respectively. There was a significant positive correlation between an increase in CTC levels and tumor vascular density (*p*-value < 0.01). This correlation was stronger than that between CTC levels and tumor size (*p*-value = 0.076). The captured CTCs were implanted into a non-tumor-bearing NTTg mouse for xenografting, confirming their viability and tumorigenesis. Conclusion: Serial CTCs during an early stage of tumor progression were quantified and found to be positively correlated with the later tumor vascular density and size. Furthermore, the successful generation of CTC-derived xenografts indicates the tumorigenicity of this early onset CTC population.

## 1. Introduction

Cancer cells detaching from the primary tumor site into circulation and being transported as circulating tumor cells (CTCs) to the distal organs comprise the initial step of metastasis. The correlation between CTCs and the incidence of metastases has long been recognized [1]. The development of CTC isolation techniques—notably, CellSearch, the only FDA-clearance technique—has greatly increased research into the clinical utility of CTCs in recent decades. An updated review has summarized evidence relating to CTCs as independent prognostic factors in several cancer types, and their potential applications in monitoring treatment [2]. However, the initial onset of CTC existence and their ability to progress to tumorigenesis during tumor development and progression remains under question. In breast cancer, the presence of cytokeratin positive disseminated tumor cells (DTCs) in the bone marrow (BM) is an independent prognostic factor in early breast cancer patients [3,4,5]. DTCs in bone marrow indicate that CTCs came from a primary tumor and settled in the bone marrow [3]. This finding implies that CTCs are present in patients with early-stage cancers. However, the reported detection rate of CTCs was only around 23% [6,7,8], which would limit its clinical utility in the early cancer stages. CTC enumeration alone is also not sufficient to address CTCs’ tumorigenesis in the early onset of neoplasm.

Current screening for breast cancer is mostly undertaken by imaging, such as through mammography or breast ultrasonography. Although screening by mammography has successfully reduced breast cancer mortality, its limitation in the detection of invasive cancer means the diagnosis rate has remained unchanged in the advanced stages [9]. Ultrasonography is recommended as an adjunctive screening tool to mammography, especially in women with dense breasts. However, its benefit in mortality reduction has yet to be proved [10,11]. Therefore, methods for the early detection of invasive cancer cells are crucial.

The question remains, how early will cancer cells be shed from the primary tumor site into circulation during tumor progression? Human clinical trials are not easy to carry out in order to answer this question, because they require a large study population and an extensive evaluation period before tumor development. Moreover, the rarity of CTCs requires highly sensitive CTC detection devices, but research into this technology remains a challenge [12,13]. Using a transgenic BALB-NeuT female mouse, which developed breast tumors spontaneously at the mammary gland, Hüsemann et al. found that DTCs become detectable in the BM in the premalignant phase of tumor development, and supportive evidence suggests they then grow into metastasis [14]. Therefore, CTCs can theoretically occur in the early stage of tumor development, but this is yet to be proven by experimental data. Previously, we developed a microfluidic chip, called the CMx chip, constructed by an anti-EpCAM-functionalized supported lipid bilayer (SLB), to capture CTCs with a high capture efficiency of up to 95% and viability of 86% after subsequently isolated elution [15,16]. We successfully applied this technology to predict cancer recurrence in non-metastasis colorectal cancer after curative resection [17], and in an assay for the detection of colorectal adenoma and cancer [18].

Here, we conduct a longitudinal, clinically uninterrupted study for CTC monitoring using an animal model. The NTTg mouse, a transgenic animal model carrying the unactivated rat HER2 (Human Epidermal Receptor 2)/neu protooncogene (NeuN) and the mouse mammary tumor virus (MMTV) promoter, presents a similar disease phenomenon to breast cancer and has a prolonged latency of tumorigenesis and growth, with late occupancy of metastasis [19,20]. The challenge of using a mouse model is that the average mouse has only ~1.5 mL of blood, and the blood required for a CTC analytical system would potentially cause animal death at the first examination. For example, a recent CTC study with a xenograft mouse model took an average of 250 μL whole blood for examination, a volume which corresponds to about one-sixth of the total blood volume of a mouse [21]. Losing this much blood will cause bodyweight loss and lead to experimental bias, possibly due to tumor development interference.

In this experiment, we successfully used only 75 μL of blood at each time point for CTC detection, maintaining the healthy status of the mice during serial blood sampling. We evaluated the occurrence of CTCs and their relation to tumor volume and vascularity using a high-resolution ultrasonography system with Doppler capability, which was able to construct a 3D model of the tumor to precisely measure the tumor interstitial tissue volume and vascularity. This is the first study to evaluate the dynamic changes of intact CTCs during a visible tumor’s development.

## 2. Materials and Methods

### 2.1. Animal Model

The NeuN transgenic (NTTg) mouse is an MMTV-promoted, unactivated–HER2/neu transgenic animal model [19]. NTTg mice have a higher mean background CTC burden than wild type Balb/c mice. According to the animal facility guidelines, the NTTg mice were maintained in the animal facility at the Genomics Research Center, Academia Sinica. All the animal experiments were performed according to the Institutional Animal Care and Use Committee (IACUC) regulations of Academia Sinica (Protocol ID: 11-10-236). The mice were screened for mammary masses every 8–12 days by manual palpation from 5 months of age.

### 2.2. Anesthesia

The anesthesia of mice was performed with 5% isoflurane in pure oxygen from a Matrix vaporizer and was maintained at 1.5–2.5% isoflurane. DuraTears (Alcon) was applied to both eyes. The hair above the treatment site was removed using a hair clipper and mild depilatory cream. After hair removal, the mice were placed on an electrical heating pad, and the surgical site was scrubbed three times with povidone-iodine and 70% alcohol swabs.

### 2.3. Measurement of Tumor Size and Vascular Density by Doppler Ultrasonography

When the tumor was palpable and had grown for one week, the tumor size and vascular density were measured by the VisualSonic Vevo 770 high-resolution image system (Toronto, ON, Canada), with a high-frequency transducer (RMV-704). After anesthesia of the mice and the application of the transducer gel to the skin, the transducer was placed in the automatic linear motor holder for the examination, with a frequency range of 20–60 MHz and a focus depth of 6 mm. The images were acquired every 0.1 mm with the Doppler set. We used partial sampling (~1 mm^3^) to calculate the vascularity, to avoid errors caused by the vibration of respiration.

### 2.4. Lumpectomy for Primary Culture of Tumor Cells and Pathology Examination

A lumpectomy was performed on the mice under anesthesia. After disinfection of the skin, an incision of the skin and dissection of the tumor from the surrounding tissue was performed using mosquito clamps and electric cauterization of bleeding vessels. The operative wound was then cleaned with a solution containing 100 mg/mL cephazolin after resection of the tumor. The wound was closed with 5-0 nylon sutures and covered by gauze with gentamycin and metronidazole ointment. The mice recovered on the heating pad with an oxygen supply.

The primary culture of tumor cells was performed according to the previous study [22]. Briefly, the tumor tissue was chopped into 3–4 mm pieces, washed by phosphate-buffered saline solution (PBS), and then digested by 0.25% trypsin-ethylenediaminetetraacetic acid (Trypsin-EDTA (1×), phenol red, Gibco, Thermo Fisher Scientific, Waltham, MA, USA) at 4 °C overnight for 16 h. The trypsin solution was then removed, and the tissue was incubated in the residual trypsin at 37 °C for 30 min. DMEM (Dulbecco modified Eagle medium) complete medium (supplemented with 5% fetal calf serum, penicillin-streptomycin, and sodium pyruvate) was then added, and the tissue was pipetted up and down to disperse it. Large pieces of undispersed tissue were allowed to settle, and the cell suspension was pipetted away from these pieces. The cells were counted and seeded into tissue culture flasks for culture in DMEM complete medium. Several rounds of differential trypsinization were performed to deplete the fibroblasts during culture, to enrich more adherent epithelial cells.

The primary cultured tumor cells were trypsinized and placed on a porous polycarbonate membrane (2 μm pore in diameter). Between each process, the cells on the membrane were washed three times with PBS for 5 min. The cells were then fixed by 4% paraformaldehyde for 10 min, and blocked by 5% bovine serum albumin (BSA, Sigma-Aldrich, Saint Louis, MO, USA) in PBS for 1 h at ambient temperature. After the blocking step, the cells were stained by the primary antibodies against CD45 (ab25052, Abcam, Cambridge, UK), HER2 (ab2428, Abcam, Cambridge, UK), and PanCK (BP5069, Acris, Herford, NRW, Germany) at 4 °C overnight, and incubated in secondary antibodies of Alexa488 (Invitrogen, Thermo Fisher Scientific, Waltham, MA, USA), Alexa568 (Invitrogen), and Cy5 (ab102372, Abcam) at ambient temperature undercover for 1 h, respectively. DAPI (4′,6-diamidine-2′-phenylindole dihydrochloride, Thermo Fisher Scientific, Waltham, MA, USA) was used for nucleus staining. The filter membrane containing the cells was fixed with a mounting medium and covered by a coverslip. For the pathological analysis of a tumor, the tumor tissue was trimmed and embedded into CBT, and stored at −80 °C. A cryosection was performed within 1 week by rotary microtome at 3 mm thickness. The staining protocol was as described in the previous section.

### 2.5. Quartz Crystal Microbalance with Dissipation (QCM-D) Measurements for Verification of Surface Modification

The silicon oxide (SiO2) coated QCM crystal chips (AT-cut quartz crystals, f0 = 5 MHz, Q-Sense AB) were cleaned in 0.1 M sodium dodecyl sulfate, followed by rinsing with Milli-Q water, drying under nitrogen, and exposure to oxygen plasma for 10 min. The concentration and the washing conditions of each coating step in the QCM-D chamber were described in Result 3.1.1. 1-Palmitoyl-2-oleoyl-*sn*-glycero-3-phosphocholine (POPC) and 1,2-dipalmitoyl-*sn*-glycero-3-phosphoethanolamine-N-cap-biotinyl (b-PE) were purchased from Avanti Polar Liquids. For QCM-D measurement, the chamber was temperature-stabilized to 25 °C. All measurements were recorded at the third overtone (15 MHz), and the data shown here were normalized to the fundamental frequency (5 MHz) by dividing the overtone number.

### 2.6. mCMx Chip Fabrication

The mCMx chip used in this experiment was slightly modified from the CMx chip [15]. The raw materials, including PMMA and the bonding sticker (60 mm thick), were engraved into a specific shape with CO2 laser engraver patterns. Before assembly, 60 mm × 24 mm coverslips and all the PMMA parts, including the nuts and engraved plates, were washed with detergent and double-distilled water in a sonicator, and dried with an air blower. The PMMA parts were then adhered together with chloroform.

The coverslips were cleaned by oxygen plasma and bonded together with PMMA by bonding stickers. The channels formed by the bonding stickers in the chips were flushed with double-distilled water and filled with PBS. Afterward, the procedures of glass surface modification were modified from the previous study by coating with biotinylated anti-mouse-EpCAM IgG (13-5791-82, eBioscience, Thermo Fisher Scientific, Waltham, MA, USA) (mCMx-EpCAM chip) or biotinylated anti-rat-HER2 IgG (610161, BD Bioscience, San Jose, CA) (mCMx-HER2 chip). The recombinant EpCAM protein (Cat#50591-M08H, Sino Biological, Beijing, China) was used for QCM analytical validation, to determine the IgG-antigen binding ratio on the mCMx chip.

### 2.7. CTCs Captured by the mCMx Chip and Immunostaining for Enumeration

A total of 75 μL of the mouse’s peripheral blood was collected by retro-orbital venipuncture with a heparin-coated capillary tube and was diluted by 375 μL EDTA-PBS at 1.8 mg/mL as the final concentration of EDTA. The 450 μL specimen was added to the reservoir and run through the mCMx chip at a 1.5 mL/h flow rate. Subsequently, 400 μL PBS was run through the chip at the same flow rate. Then, 1 mL of PBS was run through the chip at a flow rate of 9 mL/h, leading to 1.8 mL of 5% BSA-PBS foam (BSA foam) being generated by the vortex. Then, 1.8 mL BSA foam as an eluent was run through the chip at a 9 mL/h flow rate and was collected in a 1.5 mL Eppendorf tube. The eluent was dripped onto the porous polycarbonate membrane (2 μm pore in diameter). The cells were stained with mouse anti-mouse CD45 (ab25052, Abcam) and guinea-pig anti-mouse pan-cytokeratin (PanCK, BP5069, Acris) antibodies, followed by DAPI (Sigma-Aldrich) nuclear staining. Secondary antibodies, including goat anti-mouse AlexaFluor488 (Invitrogen) and goat-anti-guineapig Cy5 (ab102372, Abcam), were used for immunofluorescence staining. The sample image acquisition was undertaken with a Leica DMI600 B, 10× objective, 10× eyepieces, Hamamatsu ORCA-03G CCD camera, and the Metamorph image system.

### 2.8. Inoculation of Captured CTCs for the Tumorigenesis Test

A total of 150 μL whole blood collected from each tumor-bearing NTTg mouse (DT1, DT2, DT3, DT4, 14 months old) was run through four mCMx-EpCAM chips (10% antibiotics rinsed), respectively. The captured cells were released into an Eppendorf tube with Ham’s F-12 nutrient mix (F-12, 20% fetal bovine serum and 1% antibiotics, Gibco), and then incubated at 37 °C for 1 h. Then, the Eppendorf tube was centrifuged at 1200 rpm for 5 min, and the 1000 μL supernatant was removed carefully. The remaining eluent (500 μL) was pipetted gently and collected into a 1 mL syringe. The eluent in the syringe was inoculated subcutaneously between the scapular regions of a 14-month-old non-tumor-bearing NTTg mouse with a 26 G needle, and 0.5 mL of F12 medium was also inoculated 3 cm below the first site as a negative control.

## 3. Results

### 3.1. Modified CMx (mCMx) Chip Construction and Feasibility

We monitored the dynamic changes of the CTC count in the transgenic NTTg mouse model by using an mCMx microfluidic channel etched into a chip, a schematic of which is shown in Figure 1. The mCMx chip was designed as shown previously [15], but the coating of the inside walls of the mCMx channel was modified to make it amenable to capturing mouse CTCs. As before, the coating started with a biotinylated supported lipid bilayer (bSLB) before incubation with NeutrAvidin™ (NA) biotin-binding protein. Following the addition of NA, the immobilized anti-mouse-EpCAM (anti-mEpCAM) layer was formed by coating with either biotinylated anti-mEpCAM IgG (mCMx-mEpCAM chip) or biotinylated anti-rat-HER2 IgG (mCMx-HER2 chip).

To test the overall performance of the mCMx chips, we ran a 2 mL culture medium spiked with the pre-stained (CellTracker, Invitrogen, Waltham, MA, USA) primary cultured cells from an NTTg mouse primary tumor through each of the mCMx-mEpCAM chips or mCMx-HER2 chips. The cell recovery ratio from the mCMx-HER2 chip was 12% (34/282 on average, *n* = 3), while from the mCMx-mEpCAM chip it was 71% (220/307 on average, *n* = 3). Based on the low recovery rate of the mCMx-HER2 chip, we focused solely on the mCMx-mEpCAM chip for the rest of this study.

#### 3.1.1. Surface Coating of the mCMx Chip

The surface coating and following conjugations, including capturing antibodies and antigens, were monitored with a quartz crystal microbalance with dissipation (QCM-D). For this work, the antigen we used was anti-mEpCAM IgG. As shown in Figure 2, the QCM-D response represents a series of coatings for the coating on a SiO2-pretreated quartz crystal. First, 0.25 mg/mL of lipid vesicle solution (in phosphate buffer) containing POPC and b-PE solution (15 mol%) was dispensed into the QCM-D chamber at the point I. The normalized average frequency change ΔF, 25.0 ± 2 Hz, was characteristic of a highly uniform lipid bilayer. After two buffer washes with phosphate-buffered solution saline (PBS, Gibco, washes denoted by * in Figure 2), 0.1 mg/mL solution of NA was dispensed at point II. NA binding saturated at ΔF = 107.6 ± 6.6 Hz. At point III, 65 ug/mL of biotinylated anti-mEpCAM IgG solution was dispensed into the chambers, and binding showed ΔF = 43.5 ± 10.6 Hz. Finally, we confirmed the conjugation of 65 ug/mL antigen mEpCAM with the surface-bound anti-mEpCAM IgG at point IV, where ΔF = 18.7 Hz. These results demonstrated that anti-mEpCAM could be immobilized on an SLB in a manner analogous to previous work with anti-EpCAM-functionalized SLBs [15], and with a comparable quantity of mEpCAM on the surface.

#### 3.1.2. Cell Identification

In their previous work, Yu et al. defined CTCs as cells that were positive on pan-cytokeratin (panCK, containing CK8, 18, 19) and DAPI, but negative on CD45 [13]. In the MMTV-*neu* animal model, each cell derived from mammary tissue was anticipated to overexpress HER2 [19]. To verify the HER2 expression and mammary origin of the cells, panCK, CD45, and HER2 antibodies were used for concomitant staining of the tumors derived from NTTg mice (Appendix A). Both tissue hematoxylin–eosin staining, and immunostaining results showed a positive expression of HER2 and panCK in the mammary tumor tissue of the mice. Similarly, the positive expression of both PanCK and HER2 was also identified on the isolated tissue derived from the NTTg (Appendix A). For CTC enumeration, we adapted Yu et al.’s CTC definition; that is, panCK+/CD45−/DAPI+ cells with a round nucleus (diameter ~8–12 um), a round cell shape (diameter ~12–25 um), and a high nucleus-cytoplasm volume (N/C) ratio [23] (Figure 3A). Those with a low N/C ratio were excluded from CTC enumeration (Figure 3B). The PanCK−/CD45+/DAPI+ cells were stratified as blood-derived leukocytes (Figure 3C,D). CTC clusters were also occasionally found in the samples but were excluded from CTC enumeration (Figure 3E).

### 3.2. CTC in the Early Stage of Tumor Development and during Tumor Progression

The CTC count was determined as a function of time in two different groups of NTTg mice. The CTC counts in the first group of five mice were enumerated when the mice were at the ages of 8, 12, 16, and 20 weeks. The average CTC counts ± standard deviation during this time were 0, 1.2 ± 1.6, 0.8 ± 0.8, and 1.2 ± 0.8 cells per 75 μL blood, respectively (Figure 4A). There was no statistically significant variation in the average CTC count over this period.

To limit the volume of blood drawn from the mice, a second set of four mice (referred as DT1, DT2, DT3, and DT4) were evaluated. An initial sample of blood was taken at age 32 weeks, and then only when the tumor was palpable was a second sample taken. At 32 weeks, the average CTC count was 16 ± 9.5 cells per 75 μL blood. This higher count, compared to the mice in the first group, was determined even though the breast tumor remained undetectable by palpation or ultrasonography at that time. Subsequent monitoring found that the tumors became palpable within an additional 17 to 83 days (Appendix A). After the tumor was palpable, the CTC number was enumerated again after one week, as shown in Figure 4B. When the tumors became palpable, the average CTC count increased to 29.0 ± 18.2 cells per 75 μL blood. After one additional week, the CTC count increased further to 70.0 ± 30.3 cells per 75 μL blood.

In addition to enumerating the CTC count of the NTTg mice, once the tumor became palpable, we measured the tumor size and vascularity every week with ultrasonography with Doppler capabilities. The use of the ultrasonography system enabled us to monitor tumor angiogenesis and density, without needing to sacrifice the mice. To measure the volume of the tumor tissue, we reconstructed a stereo model of the tumor by drawing the contour manually to exclude the mucinous pool (hypoechoic area), and then calculated the volume as shown in Figure 5A,B. We determined the vascularity of the tumors with the embedded Doppler function of the ultrasonography equipment. The function was verified by scanning a normal lymph node (Figure 5C). In this case, we were able to recognize efferent vessels (200 μm in diameter) [24].

At the time the tumors became palpable, the average tumor size and vascular density in the NTTg mice were 130.8 ± 113.7 mm^3^ and 2.9 ± 1.1%, respectively. After one week, these numbers increased to 232.4 ± 188.8 mm^3^ and 5.0 ± 1.3%. The CTC number, tumor size, and vascular density increased significantly with the time of tumor progression (general univariate linear model, *p* < 0.05, Figure 5D,E). However, the correlation of CTCs to vascular density was more significant than to tumor size (Spearman’s *r* coefficient = 0.715, *p* < 0.01 for vascular density versus *r* = 0.53, *p* = 0.076 for tumor size). Despite the correlation between the increase in the CTC count and the increase in either vascularity or tumor size in the respective individuals, we found that the CTC count did not correlate with tumor volume absolutely, meaning that we cannot predict the actual tumor volume of any individual based solely on their CTC count.

### 3.3. Replanting the CTCs and Tumorigenesis

To confirm the tumorigenesis of the captured CTCs, we inoculated captured CTCs from 600 μL NTTg mice whole blood (150 μL blood from each of the four tumor-bearing mice) into one non-tumor-bearing mouse. Considering the cell viability after elution, we modified the elution rate from 9 mL/h to 6 mL/h. Three weeks after transplantation, we found a solid mass about 1 cm away from the inoculation site, and no mass was found at the control site (Figure 6).

## 4. Discussion

Our results revealed that viable CTCs are detectable early, before the time of detecting tumors by palpation. Furthermore, once the tumor became detectable, the CTC count increased in individual specimens with a positive relation to the vascularity and size of the tumor. These findings imply that CTCs exist early, and account for cell proliferation and angiogenesis of tumors.

How large does a tumor need to be for CTCs to become detectable? When a tumor grows beyond 2 mm^3^, it can undergo angiogenesis [25], at which point it will have the potential to release tumor cells into new blood vessels. In our experiments, CTC counts fluctuated in a low range while the tumor was smaller than ~50 mm; however, once the tumor was larger than ~50 mm the CTC count grew and continued to increase with tumor growth and vascularity. Other experiments with the lung metastasis model counted CTC numbers obtained from mice with tumor weights between 0.02 to 5.96 g, but this experiment had to sacrifice twelve mice every ten days between day 15 and 55 to obtain these different tumor sizes [26]. In contrast, our experiment only required drawing 75 μL of blood from mice to isolate the CTCs, thereby allowing us to track the dynamic relationship between CTC numbers and tumor growth throughout the same mouse’s lifetime.

Our results showed that CTC counts are more strongly correlated with tumor vascularity than tumor size (*p* < 0.01 for vascular density versus *p* = 0.076 for tumor size). Angiogenesis can provide enough blood flow and nutrition to rapidly growing tumor tissue. When vessels are proliferating, it is easy for their structure to develop imperfections, such as a looser endothelium [27]. Higher vascularity provides a larger endothelial surface area to interact with tumor cells [28], and can also increase tumor cells’ intravasation rates [29]. The above conditions make it easier for tumor cells to disseminate into blood vessels. Clinically, vascular invasion has been shown to provide an independent prognosis of the oncologic outcome of patients with primary breast carcinoma [30]. One clinical study reported that, in early breast cancer, CTCs were found in 13 of 48 pre-surgery patients. These 13 patients had higher rates of negative prognostic features, i.e., high proliferation, large tumor dimension, lymph node positivity, and negative receptor status, compared to the other patients. In particular, vascular invasion showed a statistically significant correlation with CTC-positivity [31]. These results suggest that CTCs may be linked to vascular invasion, and to other known negative prognostic factors.

One additional contribution of our study was the successful implementation of a CTC-derived xenograft. This helps explain the tumor recurrence of some patients with early cancer after curative resection, and points to the rationality of applying CTC technology to the screening of patients at a high risk of tumor development in future medical research.

The low counts of CTCs in patients with early stage tumors requires sensitive CTC assays and analysis of large volumes of blood [32]. In addition to the technical challenges, detecting CTCs in human patients before development of a tumor would require prohibitively large study populations and long evaluation times [33]. One approach to circumvent these real-world challenges is to focus on patients with a high risk of developing cancer or metastasis. For example, a study has reported that CTCs could be detected in 3% of 168 patients with chronic obstructive pulmonary disease (COPD), and annual surveillance of these CTC-positive COPD patients showed lung nodules on a CT scan one to four years after CTC detection, leading to prompt surgical resection and histopathologic diagnosis of early-stage lung cancer. In contrast, no CTCs were detected in the control smoking and healthy nonsmoking individuals [34]. Another multicenter prospective study using a filtration-based detection technique (ISET: Isolation by Size of Epithelial Tumor Cells) was conducted, with three CTC screening rounds in 614 eligible patients at one-year intervals for lung cancer screening [35]. The results revealed that the sensitivity of baseline CTC detection for lung cancer detection was 26% (95% CI 12–49), and the performance was insufficient to predict subsequent development of lung cancer (2 of the 13 lung cancers and 0 of 13 extrapulmonary cancers) during the two-year follow-up period. In our previous study, which used a CMx CTC assay for 667 patients, the results showed that precancerous lesions of the colon (76.6% sensitivity, 97.3% specificity) could be detected with accuracy as a malignant lesion of the colon (87% sensitivity, 97.3% specificity) [36], more effectively that with the current fecal immunochemical test (50% sensitivity, 97.3% specificity) [37].

Patients with non-metastasis cancer also risk developing recurrence during follow-up after curative resection of the tumor. There is an increasing number of publications on patients with an earlier stage cancer that demonstrate significant correlations between CTC counts and metastasis relapses in patients with several kinds of tumor, such as breast [6,7], colorectal [17], bladder [38], liver [39], and esophageal cancer [40]. A recent review also reported that CTCs can serve as a prognostic biomarker factor for breast cancer [41]. However, clinical application of CTC technology based on only enumeration has its limitations. It is estimated that aggressive tumors release thousands of CTCs each day, but only 0.1% of CTCs survive the various stress factors and form distant metastases [42,43]. The present CTC technology also cannot identify all subtypes of CTCs. For example, in a single CTC analysis study of advanced estrogen-receptor (ER)-positive/human epidermal growth factor receptor 2 (HER2)-negative breast cancer, 84% of patients had CTCs expressing HER2-positive and cultured HER2+, and HER2− CTCs spontaneously interconverted their phenotype according to flow cytometry identification [44].

We used the mEpCAM antibody to capture CTCs in this experiment. We also tried to use mHER2 instead of mEpCAM antibodies for binding and capturing the CTCs from the breast tumors of the NTTg mice, however this was unsuccessful. On the other hand, the mHER2-based microfluidic device has successfully isolated the CTCs of patients with metastatic breast and gastric cancers [45]. Because the transgene is rat HER2/neu in NTTg mice [19], we tried several different clones of anti-human-HER2 IgG by immunostaining, and most were good for immunostaining but poor for capturing on our platform. The capture efficiency using mHER2 was around 12%, compared to 71% using the mEpCAM antibody. An improvement of the mHER2-based captured capture efficiency is needed in the future.

While anti-mHER2 was not optimized to capture CTCs, it was used for immunostaining to specify the molecular characteristics of CTCs. We confirmed the positive expression of HER2 in PanCK+/DAPI+ NTTg-derived CTCs (Appendix A). Apart from the classical definition of CTCs, namely PanCK+/CD45−/DAPI+ cells (Figure 3), we also found some subtypes that were HER2+ only, PanCK+ only, as well as HER2−/ PanCK−. Other studies have suggested that the appearance of CK+ only cells in the MMTV-NeuT transgenic mouse model might be due to chaotic genetic expression caused by an aberrant gene [14], or might represent contaminants from the skin or mucosa during blood sampling. Some researchers regard HER2+ only cells as the result of epithelial-to-mesenchymal transition, and HER2−/PanCK− cells as arising from the endothelium or as the result of epithelial-to-mesenchymal transition [46]. In addition to single CTCs, we observed CTC microemboli occasionally (Figure 3E). Clinical research has revealed that CTC microemboli are related to poor prognosis or advanced disease in breast, lung, pancreatic, and colorectal cancer [16,47,48,49].

For patients with unilateral breast cancer, contralateral prophylaxis mastectomy is increasingly being utilized, especially among young patients [50,51]. However, the benefit of prophylaxis mastectomy has only been proven in some specific genetic-related cancer types, such as BRCA1 and BRCA2, in reducing contralateral cancer risk; the benefit for survival remains under debate [52,53]. Others situations, such as young patients, minor genetic mutations (CHEK2, TP53, ATM, PALB2, PTEN, CDH1), or family history without known gene mutation, have shown no evidence of any benefits of prophylaxis mastectomy [54]. Nevertheless, prophylaxis mastectomy is still being used with increasing preference in non-BRCA patients due to low breast satisfaction levels, fear of recurrence, and cosmetic symmetry [55,56]. A recent study reported that *PIK3CA* hotspot mutations were detected in 37/56 (66.1%) and 23/27 (85.2%) of DNA samples isolated from CellSearch^®^ cartridges in early and metastatic breast cancer, respectively [57]. *PIK3CA* hotspot mutations have been reported to confer in vitro and in vivo tumorigenicity [58]. Metastatic breast cancer-derived CTCs were used to generate mice xenografts for a possible drug sensitivity test in a proof-of-concept study. A genome sequencing of the CTC lines revealed pre-existing mutations in the *PIK3CA* gene, and other newly acquired mutations [59]. However, generating patient-derived xenograft models for clinical use is impractical because it is time-consuming, laborious, and has a low success rate. Combining molecular markers related to tumorigenesis with a series of viable CTC retrievals may offer an opportunity to conduct an early evaluation of prophylaxis mastectomy. Such an evaluation technique could be developed from our research correlating CTC counts with tumor growth in mice.

## 5. Conclusions

CTCs are detectable at the early stage of breast tumor development and are positively correlated with tumor vascularity and tumor size. The generation of CTC-derived xenografts further confirmed the tumorgenicity of these early onset CTCs. This study demonstrates that isolating viable CTCs for further culture and molecular analysis, in addition to counting CTCs, is a promising approach to study early tumor development and preventive treatment. 

## Figures and Tables

**Figure 1 cancers-13-03294-f001:**
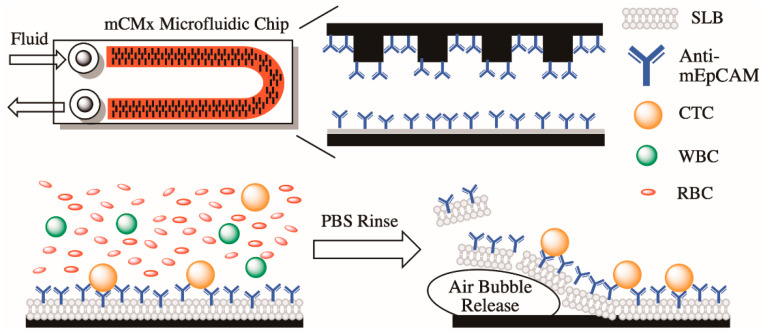
Schematic of the capture and subsequent recovery of circulating tumor cells with the mCMx system. Samples of 75 μL of mouse blood were diluted to 450 μL and injected into the microfluidic channel of the mCMx system. Circulating tumor cells (CTCs) bound to the binding sites of the anti-mEpCAM-conjugated, supported lipid bilayer (SLB) surfaces. After a phosphate buffered saline (PBS) rinse to remove non-specific binding cells, the CTCs remained adsorbed in the mCMx system. Subsequent purging with air foam disintegrated the SLB, thereby releasing the CTCs for further immunostaining or cultivation.

**Figure 2 cancers-13-03294-f002:**
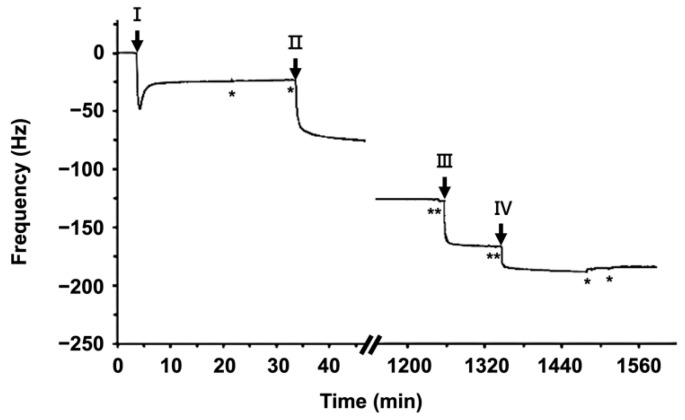
Surface modification and the non-fouling effect monitored by QCM-D. The kinetics of the stepwise formation of the surface coating, monitored by normalized frequency F using QCM-D for coating with mEpCAM antigens. Lipid vesicles containing 4 mol% of b-PE were introduced at point (I) and absorbed and ruptured to form bSLB on a silicon dioxide chip. NeutrAvidin (NA) solution was injected at point (II) and incubated for 20 h for specific binding on SLB, forming NA-bSLB. Biotinylated anti-mEpCAM IgG solution was added at point (III) to conjugate anti-mEpCAM to the NA-bSLB layer. Then, recombinant mEpCAM was injected at point (IV) to determine the IgG-antigen binding ratio. The pH 7.2 PBS buffer rinse, denoted as *, was performed at least once after each step.

**Figure 3 cancers-13-03294-f003:**
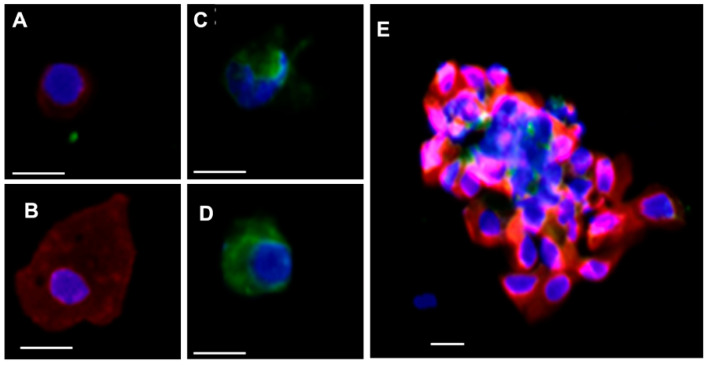
The immunostaining and morphology of captured cells by the mCMx platform. The cells were immunostained with CD45 (green), PanCK (red), and DAPI (blue) nuclear stain. (**A**) The PanCK+/CD45−/DAPI+ cells were identified as CTCs. (**B**) The PanCK+/CD45−/DAPI+ cells with low N/C-ratio were not counted as CTCs. (**C**,**D**) The PanCK−/CD45+/DAPI+ cells were defined as leukocytes. (**E**) Multiple cell clusters containing PanCK+/DAPI+ and or CD45+/− were defined as CTC clusters or CTC microemboli (all the images were acquired by Leica DMI6000 B. Scale bar = 10 μm).

**Figure 4 cancers-13-03294-f004:**
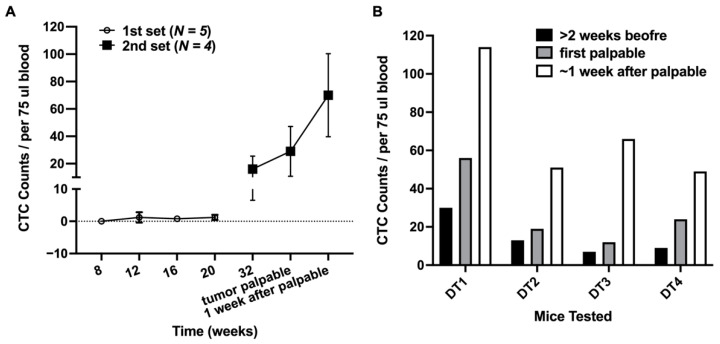
CTC counts in NTTg mice over time. (**A**) circles: Average CTC enumeration at 8, 12, 16, and 20 weeks of age in the first set of five mice; squares: Average CTC enumeration at 32 weeks of age, the week when the tumor was first detected by palpation, and approximately one week after the tumor was detected in the second set of four mice. (**B**) CTC enumeration of each of the second set of four mice (DT1–DT4) prior to, at the time of, and after the tumor became palpable.

**Figure 5 cancers-13-03294-f005:**
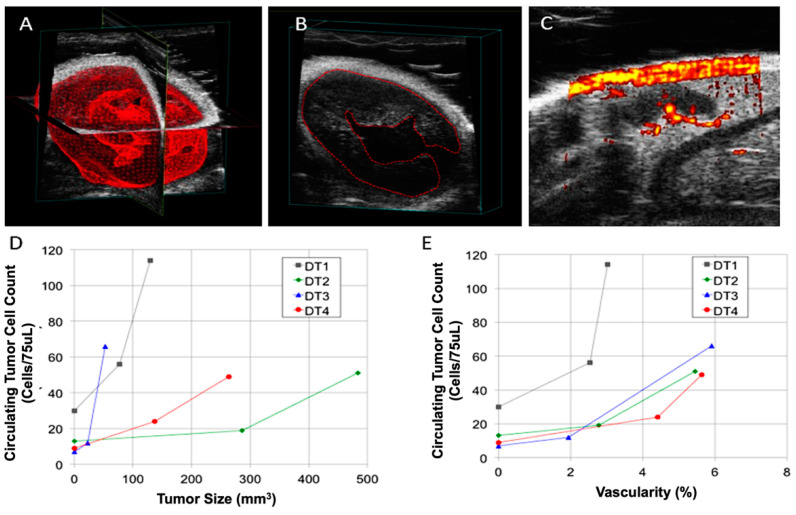
Evaluation of tumor size and vascularity in NTTg mice over time, and correlation with CTC count. (**A**) A reconstructed 3D model of a tumor acquired from an ultrasound scan. (**B**) The volume of the tumor was determined by outlining its contours in the scan. The outline of the tumor includes the mucinous pool. (**C**) Vascularity of a lymph node, as determined by an embedded Doppler function in the ultrasound scan. (**D**) The correlation between tumor size and CTC count. (**E**) The correlation between vascularity and CTC count. The lines in (**D**, **E**) are provided only to guide the eye.

**Figure 6 cancers-13-03294-f006:**
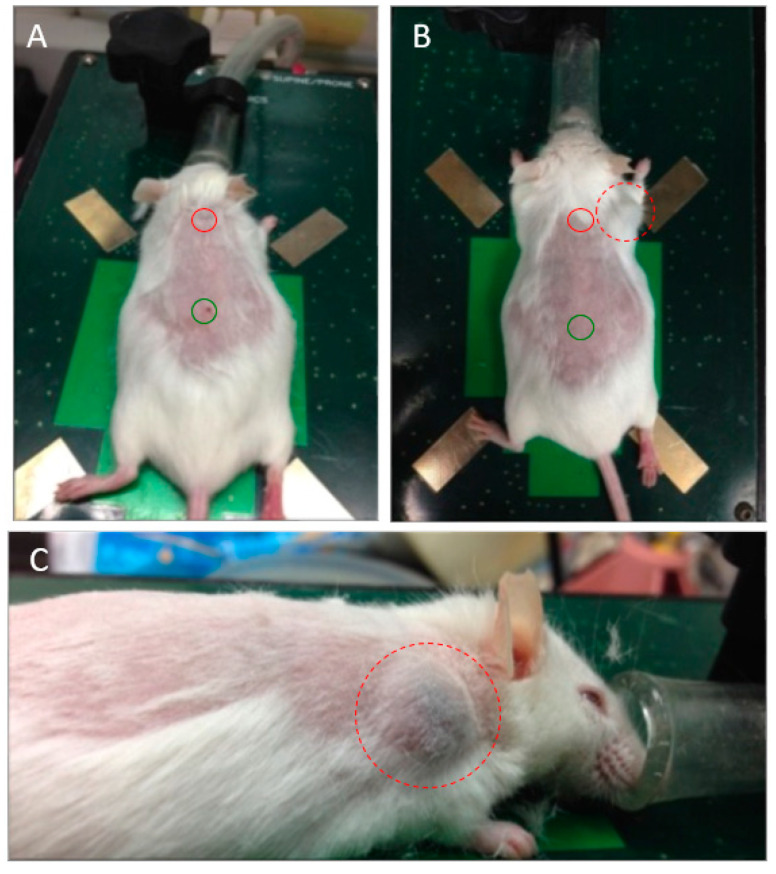
Tumorigenesis after injection of captured CTCs. (**A**) Eluted CTCs from the mCMx system were subcutaneously injected into a non-tumor-bearing mouse at the site between the scapulars (red circle), and a control medium was injected 5 cm caudally (green circle). (**B**) Three weeks later, a tumor was found 1 cm from the inoculated site (dashed red circle). (**C**) Profile view of the tumor.

## Data Availability

The data presented in this study are available on request from the corresponding author.

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
