# Peer review of "Early Detection and Dynamic Changes of Circulating Tumor Cells in Transgenic NeuN Transgenic (NTTg) Mice with Spontaneous Breast Tumor Development"

_cancers, 2021, doi:10.3390/cancers13133294_

Round 1

Reviewer 1 Report

The manuscript “Early Detection and Dynamic Change of Circulating Tumor Cells in Transgenic NeuN   Transgenic (NTTg) Mouse with 3 Spontaneous Breast Tumor Development” written by Wen-Sy Tsai et al. aimed to confirm a microfluidic chip's ability on the early detection of CTCs in a murine model. The paper present interesting topics, e.g., a) the need to understand when a cancer lesion begins to spread CTCs and then trigger metastasis onset, b) the idea to study cell spreading through mouse models, and c) CTC viability and growth capabilities once sorted from patients.

However, conclusions do not always seem to be supported by data, and the manuscript quite suffers from a lack of innovative points of view due to the already published huge amount of literature in this field and from the authors themselves, including reviews.  Hence, it could be published after some revisions only.

First, the authors state “CTCs exist early”. Without mentioning the quality of English, the sentence is not innovative. As they already reported, studies on CTCs and DTCs in the clinic already addressed that cancer may spread very early. CTCs were even detected one to four years earlier than radiological signs of malignancy on computed tomography and they have been already related to vascular invasion. [for example, in addition to the literature already reported by the authors, see Vasseur A et al., Mol Oncol. 2020; Ilie M et al., Plos One, 2014; Maltoni R et al., Cancer Lett. 2015]. Moreover, they say that the technique is feasible to screen cancer clinically or that it is valuable in the clinical application of cancer surveillance and the evaluation of adjuvant chemotherapy. They should strengthen the link with their previous published work in order to confirm their statement: from the present paper it is not obvious.

Indeed, how can the feasibility and value of the CMx chip be inferred from just mouse models? Without any comparison, at least hypothesized and discussed, with the actual clinical setting it is impossible to state this conclusion. Of course, if the authors mentioned these results in their previous papers, there is no need to propose these conclusions again here.

Moreover, no comparison was even proposed with the only gold standard, the Cell Search system. Indeed, the sentence “the low detection rates (~23%) hinder its clinical use and not sufficient to address CTC's presence at the early onset of neoplasm and tumorigenesis.” is misleading, suggesting the Cell Search is not helpful in monitoring CTCs.  If the authors already reached these results and published them (e.g., ref. 14 of the paper), the herein presented work could be more oriented towards the preclinical investigation of early CTC spreading. For example, it could have been very interesting to analyze the molecular characteristics of the cells in the primary tumor, of those detected during tumor progression, and even of those re-implanted and grown. This could have shed some light on the molecular mechanism of CTC spreading and it could have skyrocketed manuscript importance. The authors stated CTCs were viable and capable to give birth to other tumors. What about RNASeq analysis of those CTCs? At least, discuss these possibilities.  As it is, the paper is one among many other CTC-chip/methods principally able to enumerate CTCs only.

Minor revisions

Revise English

Fig 1 is a partial reproduction of author’s previous work. It is superfluous or it has to be modified to clarify elements of difference.

Fig 6 is not worthwhile to be so large. It should be reduced or modified.

Author Response

We are grateful for your efforts in providing us invaluable comments. Please see the responses and revised manuscript in the attachment, thank you!

Reviewer 2 Report

This paper reports on a CMx chip slightly modified to capture mouse CTCs (via murine EpCAM). The authors NeuN transgenic investigated with their device mice with spontaneous breast tumor development to evaluate the dynamic change of CTCs during tumor development. 75 μL of peripheral blood (peri-orbital) was collected from age of 8 weeks at several time points until tumors became palpable.

CTCs counts were 1± 1.6 to 16 ± 9.5 per 75 μL of blood before tumors became palpable and were increasing after palpable tumor formation. In addition, the authors present one transplantation experiment (collected enriched blood from four mice) in which they claim tumor formation from these CTCs. According to the authors the result demonstrates “the potential of early detection of viable cancer cells in the blood for treatment evaluation”.

Major:

The chip technology is interesting but not new. Here it was applied with an anti-mouse EpCAM antibody to a few transgenic mice developing breast cancer. The idea to monitor CTC development during tumor development is also not new and early dissemination has been demonstrated already in breast cancer models, but also in PDAC and melanoma mouse models. In principle, some pilot experiments are shown here that do not justify publication in Cancers. For validation of their method it would have been more convincing if a mouse model with a reporter gene would have been used.  In such a Model, the authors could have determined the total (expected) CTC number in the whole blood at the different time points by their and and independent method (e.g. FACS). Then, they could have applied their 75 μL assay to test the performance.

Author Response

We are grateful for your efforts in providing us invaluable comments. Please see the responses and revised manuscript in the attachment. Thank you!

Reviewer 3 Report

The manuscript shows proof-of-concept for the modified CMx chip to capture mouse circulating tumor cells derived from a spontaneous NTTg breast cancer model. The method and result section provide a clear description of the modification and analytical validation of the assay. It is a well-defined small study. However, the experiments presented in this manuscript do not clearly support the aim of early cancer detection. As the sample sizes are very small, the clinical impact should be more critically discussed. The abstract and introduction would benefit by additional proofreading.

Minor remarks to the authors:

  1. Are the anti-EpCAM antibodies maintained on your cells after release from the CMx chip? How would that affect your anti-CD45 staining as both antibodies seem to be of mouse origin?
  2. Figure 3E does not show a HER2+ CK+ cell as described in line 266, but only features the CK staining
  3. The authors mention that they tested different anti-HER2 antibody clones for immunostaining (line 405), which confirmed that the primary tumor cells were HER2 positive. It would be beneficial to include this data, as it also supports the choice of anti-HER2 staining on the CTCs despite the inefficiency as a capture reagent.
  4. Line 278-283, which describes the initial mouse study, does not refer to Figure 4A, but it seems that the results are included in this figure?
  5. Figure 4A: It would perhaps be more suitable to divide the two studies into separate graphs or at least the two studies and sample sizes should be mentioned in the figure text.
  6. The time between week 32 and palpable tumor range from 17 to 83 days. In order to support the early appearance and detection of CTCs it would be interesting to reveal whether there was any correlation between CTC count and time to palpable tumor, eg was DT3 (with the lowest CTC count at week 32) also the mouse where you waited 83 days until next sampling?
  7. The choice of a 14-month-old NTTg mouse for the CTC inoculation seems surprising and should be clarified.
  8. Line 124: I think clamp/forceps is missing after mosquito
  9. There is also a word missing in line 142
  10. Many new abbreviations are introduced in section 2.5. Please define them.
  11. The recombinant EpCAM protein used for analytical validation in figure 2 is not described in materials and methods.
  12. There is a mistake in the text to figure 5E, which shows vascularity against CTC count and not tumor size.
  13. The experiments in this study do not show data, which support “CTCs […] accounts for cell proliferation and angiogenesis of tumor” (line 343-4)

Author Response

(The authors gave the same response as above.)

Reviewer 4 Report

Wen-Sy Tsai1 et al.  at the article presents the data of the NeuN transgenic (NTTg) mice with spontaneous breast tumor development to evaluate the dynamic change of circulating tumor cells (CTCs) during tumor development.

The authors present original data demonstrating that intact and viable CTCs are found at an early stage in breast tumor development. The increase in the number of CTCs is associated with vascularization and tumor size. The article is very interesting in its design.

While reading the article there was a number of questions. 1. In their study, the authors evaluate CTCs based on Epcam only. However, literature data show a great heterogeneity of CTCs.For example, it has been shown that in patients with early stage breast cancer T2-4N0-3M0 significant changes in the quantity of the different subsets of circulating tumor cells in patients' blood were observed after carrying out the 3rd course of NACT. NACT causes significant changes in the quantity of six CTC subsets, with various combinations of stemness and epithelial-mesenchymal transition (EMT) properties (Heterogeneity of Circulating Tumor Cells in Neoadjuvant Chemotherapy of Breast Cancer. Molecules. 2018 Mar 22; 23 (4): 727.doi: 10.3390 / molecules23040727. PMID: 29565320; PMCID: PMC6017975.)

  1. Authors write (Line 99): The NTTg mice have a higher mean background CTC burden than wild type Balb/c mice.(99) What does it mean?
  2. In Fig. 4, it is necessary to indicate the unit of measurement on the first graph along the Y axis.

Thanks to the authors for the interesting article.

Author Response

(The authors gave the same response as above.)

Round 2

Reviewer 2 Report

The authors improve their first version of the manuscript. It contains some interesting information concerning the HER2mouse and dissemination of CTCs, but a problem with over-interpretation/exaggeration and English Language persists (e.g. why repetitively "would" in the simple summary - plus several language issues in several section of the paper). 

The authors must clearly highlight that the data from a transgenic mouse model cannot be directly transferred to the human situation. There are several statements throughout the paper and I do not want to highlight all of them. But for example, already in the simple summary there is this conclusive statement  "This result confirms the feasibility of early detection of viable cancer cells in the blood for clinical applications". How can these data confirm anything for clinical applications? This is just not correct.

Also, in their conclusions the authors write :"Intact and viable CTCs with tumorigenesis are detectable at the early stage of breast tumor development by the mCMx system, and CTC numbers are positively correlated with tumor vascularity and tumor size." (also: "CTC with tumorigenesis" is incorrect English) Here, the authors insinuate that all of their "early" CTCs are tumorigenic, which was not shown. This was only demonstrated for pooled blood from mice bearing palpable tumors (which would be gigantic in humans). The problem is that such slightly twisted arguments are made throughout the paper, which creates a problem for the whole work. 

Another point relates to the specificity of the test that was questioned in the previous review but not answered by the authors. 
